# Revisiting Frank-Wolfe for Polytopes: Strict Complementarity and Sparsity

**Dan Garber**
Department of Industrial Engineering and Management
Technion - Israel Institute of Technology
Haifa, Israel 3200003
dangar@technion.ac.il

## Abstract

In recent years it was proved that simple modifications of the classical Frank-Wolfe algorithm (aka conditional gradient algorithm) for smooth convex minimization over convex and compact polytopes, converge with linear rate, assuming the objective function has the quadratic growth property. However, the rate of these methods depends explicitly on the dimension of the problem which cannot explain their empirical success for large scale problems. In this paper we first demonstrate that already for very simple problems and even when the optimal solution lies on a low-dimensional face of the polytope, such dependence on the dimension cannot be avoided in worst case. We then revisit the addition of a strict complementarity assumption already considered in Wolfe's classical book [29], and prove that under this condition, the Frank-Wolfe method with away-steps and line-search converges linearly with rate that depends explicitly only on the dimension of the optimal face. We motivate strict complementarity by proving that it implies sparsity-robustness of optimal solutions to noise.

## 1 Introduction

The Frank-Wolfe method (aka conditional gradient, see Algorithm 1 below), originally due to [8] is a classical first-order method for minimizing a smooth and convex function over a convex and compact set [8, 24, 19]. It regained significant interest in the machine learning, optimization and statistics communities in recent years, with some notable applications including structured prediction [22] and video co-localization [20], mainly due to two reasons: i) in terms of the feasible set, the method only requires access to an oracle for minimizing a linear function over the set. Such an oracle could be implemented very efficiently for many feasible sets that arise in applications, as opposed to most standard first-order methods which usually require to solve non-linear problems over the feasible set (e.g., Euclidean projection onto the set) which can be much less efficient (e.g., see detailed examples in [19, 18]), and ii) when the number of iterations is not too large, the method naturally produces sparse solutions, in the sense that they are given explicitly as a convex combination of a small number of extreme points of the feasible set, which in many cases (e.g., optimization with sparse vectors /low-rank matrices) is much desired ([19, 6]).

The convergence rate of the method is of the order $O(1/t)$ where $t$ is the iteration counter. This rate is known to be tight and does not improve even when the objective function is strongly convex, a property that, when combined with smoothness, is well known to yield a linear convergence rate, i.e., $\exp(-\Theta(t))$ for standard first-order methods such as the proximal/projected gradient methods. For optimization over convex and compact polytopes, in his classical book [29], Wolfe himself suggested a simple variant of the method that does not only add new vertices to the solution using the linear optimization oracle, but also moves away more aggressively from previously found vertices of the

polytope, a step typically referred to as an away-step. Wolfe conjectured that with the addition of these away-steps and assuming strong convexity of the objective and an additional strict complementarity condition w.r.t. the optimal face of the polytope (see Assumption 2 in the sequel), a linear convergence rate can be proved. Later, Guélat and Marcotte [17] proved this result rigorously but without giving an explicit rate or complexity analysis. Also, their convergence rate depends on the distance of the optimal solution from the boundary of the optimal face of the polytope, which can be arbitrarily bad. Their technique for proving the linear rate is also related to techniques used in [3, 11].

In recent years Garber and Hazan [10, 12] and then Lacoste-Julien and Jaggi [21] presented variants of the Frank-Wolfe method that utilize away-steps alongside new analyses, which resulted in provable and explicit linear rates without requiring strict complementarity conditions and without dependence on the location of the optimal solution. These results have encouraged much followup theoretical and empirical work e.g., [2, 26, 25, 14, 27, 13, 28, 16, 5, 15, 1, 4, 23, 7], to name a few. However, the linear convergence rates in [10, 12, 21] and follow-up works depend explicitly on the dimension of the problem (at least linear dependence, i.e., the convergence rate is of the form $\exp(-\Theta(t/d))$, where $d$ is the dimension)[1].

Unfortunately, the explicit dependence on the dimension in all such works fails to explain and support the good empirical performance of these away-steps-based variants for large-scale problems. In particular, the examples constructed to show that explicit dependence on the dimension is mandatory in general (see for instance [12]) have focused on the case that the optimal solution lies on a high-dimensional face of the polytope[2]. However, this leaves open the natural question:

*Can explicit dependence on the dimension be avoided when the set of optimal solutions lies on a low-dimensional face of the polytope?*

Indeed, models in which the optimal solution satisfies some notion of sparsity are extremely common and important in statistics and machine learning. With this respect, the solution being on a low-dimensional face is analogues to sparsity in case the feasible set is a polytope, since it implies the solution could be expressed as a small number of extreme points of the polytope.

In this work we begin by answering the above question on the negative side, at least in worst-case. We give a construction of a very simple problem for which the optimal solution is a vertex of the polytope (i.e., lies on a face of dimension 0), but for which all Frank-Wolfe-type methods (including those which use away-steps) which apply for arbitrary polytopes, require number of steps that depends explicitly on the dimension. We then revisit the strict complementarity condition assumed in the works of Wolfe [29] and Guélat and Marcotte [17] (but not in the more modern works such as Garber and Hazan [10, 12] and Lacoste-Julien and Jaggi [21]). We first motivate this condition by showing how it implies a robustness-to-noise property of optimal solutions. That is, under this condition if the optimal solutions lie on a low-dimensional face of the polytope, then also the optimal solutions to a slightly-perturbed version of the problem must also lie on this face. We also provide a simple empirical evidence for strict complementarity in a standard setup of recovering a sparse vector from (random) linear measurements. We then use this condition to give a new analysis for the Frank-Wolfe method with away-steps and line-search that converges with linear rate that depends explicitly only on the dimension of the optimal face, and not on the dimension of the problem. In terms of techniques, we use the original algorithm used in the works of Guélat and Marcotte [17] and Lacoste-Julien and Jaggi [21] (Algorithm 2 below), but with a new complexity analysis that is mostly inspired by that of Garber and Hazan [12].

It is important to note that while Garber and Meshi [13] gave a Frank-Wolfe variant for polytopes with linear rate that depends only on the dimension of the optimal face, their result can be efficiently implemented only for a very restrictive family of polytopes, and hence is far from generic. In a follow-up work by Bashiri and Zhang [1] the approach in [13] was extended to general polytopes; however, i) as in [13] it requires an oracle for computing a specialized away-step which in general is not necessarily efficient to implement (e.g., in terms of the standard linear optimization oracle), and ii) the convergence rate for general polytopes still depends on the ambient dimension of the

| ref. | quad. growth? | strict comp.? | dist$(\mathbf{x}^*, \partial\mathcal{F}^*)$? | #iterations to $\epsilon$ error |
|---|---|---|---|---|
| [17] | ✓ | ✓ | ✓ | no explicit bound proven |
| [12] | ✓ | x | x | $(\beta/\alpha)d\log 1/\epsilon$ |
| [21] | ✓ | x | x | $(\beta/\alpha)d\log 1/\epsilon$ |
| Thm. 5 | ✓ | ✓ | x | $\frac{\alpha\beta}{\delta^2 \dim \mathcal{F}^*} + \frac{\beta}{\alpha}\dim\mathcal{F}^* \log 1/\epsilon$ |

Table 1: Comparison of assumptions and convergence rates for optimization over the unit simplex. $\alpha, \beta, \dim\mathcal{F}^*, \delta$ denote the quadratic growth, smoothness, dimension of optimal face and strict complementarity, respectively. dist$(\mathbf{x}^*, \partial\mathcal{F}^*)$ denotes the distance of the optimal solution $\mathbf{x}^*$ from the boundary of the optimal face $\mathcal{F}^*$. We exclude [13, 1] since their results for general polytopes are not necessarily efficiently implementable (e.g., in terms of the standard linear optimization oracle) and the convergence rate depends on the ambient dimension $d$. For our result we assume $\dim\mathcal{F}^* > 0$ since otherwise the method has finite convergence, see Theorem 4 and Footnote 3.

polytope (and is actually potentially worse than [10, 12, 21]). On the other hand, in this work we do not impose any additional structural assumption on the feasible polytope or assume the availability of a new optimization oracle. See Table 1 for an example of comparison of related work in case the polytope is simply the unit simplex.

## 2 Preliminaries

Throughout this work we let $\|\cdot\|$ denote the standard Euclidean norm for vectors in $\mathbb{R}^d$ and the spectral norm (i.e., largest singular value) for matrices in $\mathbb{R}^{m \times d}$. We use lower-case boldface letters to denote vectors and upper-case bold-face letters to denote matrices. for a matrix $\mathbf{A} \in \mathbb{R}^{m \times n}$ we let $\mathbf{A}(i) \in \mathbb{R}^n$ denote the $i$th row of $\mathbf{A}$.

Throughout this work we consider the following convex optimization problem:

$$\min_{\mathbf{x} \in \mathcal{P}} f(\mathbf{x}), \tag{1}$$

where $\mathcal{P} \subset \mathbb{R}^d$ is a convex and compact polytope in the form $\mathcal{P} := \{\mathbf{x} \in \mathbb{R}^d \mid \mathbf{A}_1 \mathbf{x} = \mathbf{b}_1,\ \mathbf{A}_2 \mathbf{x} \leq \mathbf{b}_2\}$, $\mathbf{A}_1 \in \mathbb{R}^{m_1 \times d}$, $\mathbf{A}_2 \in \mathbb{R}^{m_2 \times d}$, $f : \mathbb{R}^d \to \mathbb{R}$ is convex and $\beta$-smooth (Lipschitz gradient). We let $\mathcal{V}$ denote the set of vertices of $\mathcal{P}$. We let $f^*$ denote the optimal value of Problem (1) and we let $\mathcal{X}^* \subseteq \mathcal{P}$ denote the set of optimal solutions. For a face $\mathcal{F}$ of $\mathcal{P}$ we define:

$$\dim\mathcal{F} := d - \dim\text{span}\{\{\mathbf{A}_1(1), \cdots \mathbf{A}_1(m_1)\} \cup \{\mathbf{A}_2(i) : i \in [m_2], \forall \mathbf{x} \in \mathcal{F} : \mathbf{A}_2(i)^\top \mathbf{x} = \mathbf{b}_2(i)\}\}.$$

We let $\mathcal{F}^* \subseteq \mathcal{P}$ denote the lowest-dimensional face of $\mathcal{P}$ containing the set of optimal solutions, i.e., $\mathcal{X}^* \subseteq \mathcal{F}^*$. In the following we write $\mathcal{F}^* = \{\mathbf{x} \in \mathbb{R}^d \mid \mathbf{A}_1^* \mathbf{x} = \mathbf{b}_1^*,\ \mathbf{A}_2^* \mathbf{x} \leq \mathbf{b}_2^*\}$. Observe that the rows of $\mathbf{A}_1^*$ are exactly the rows of $\mathbf{A}_1$ plus the rows of $\mathbf{A}_2$ which correspond to inequality constraints that are tight for all points in $\mathcal{F}^*$ and the vector $\mathbf{b}_1^*$ is defined accordingly. The rows of the matrix $\mathbf{A}_2^*$ are exactly the rows in $\mathbf{A}_2$ which correspond to inequality constraints that are satisfied by some of the points in $\mathcal{F}^*$ but not by others, and the vector $\mathbf{b}_2^*$ is defined accordingly.

We let $\mathbb{A}^*(\mathcal{P})$ denote the set of all $\dim\mathcal{F}^* \times d$ matrices whose rows are linearly independent rows chosen from the rows of $\mathbf{A}_2^*$. Similarly to [12], we define $\psi^* = \max_{\mathbf{M} \in \mathbb{A}^*(\mathcal{P})} \|\mathbf{M}\|$ and $\xi^* = \min_{\mathbf{v} \in \mathcal{V} \cap \mathcal{F}^*} \min_i \{\mathbf{b}_2^*(i) - \mathbf{A}_2^*(i)^\top \mathbf{v} \mid \mathbf{b}_2^*(i) > \mathbf{A}_2^*(i)^\top \mathbf{v}\}$ (note here $\psi^*, \xi^*$ are only defined w.r.t. the optimal face $\mathcal{F}^*$). We denote by $D$ and $D_{\mathcal{F}^*}$ the Euclidean diameter of $\mathcal{P}$ and $\mathcal{F}^*$, respectively.

Given a set $\mathcal{W} \subset \mathbb{R}^d$ we let conv$\{\mathcal{W}\}$ denote the convex-hull of the points in $\mathcal{W}$, we let nnz$(\cdot)$ denote the number of nonzero entries in a given vector, and for any positive integer $n$, we let $\mathcal{S}_n$ denote the unit simplex in $\mathbb{R}^n$. Given a point $\mathbf{x} \in \mathbb{R}^d$ and a set $\mathcal{W} \subset \mathbb{R}^d$ we denote dist$(\mathbf{x}, \mathcal{W}) = \inf_{\mathbf{y} \in \mathcal{W}} \|\mathbf{y} - \mathbf{x}\|$.

Throughout this paper, unless stated otherwise, we assume the objective function $f(\cdot)$ satisfies the quadratic growth property, which is a weaker assumption than assuming strong-convexity, and is common to all linearly-converging Frank-Wolfe variants previously studied.

**Assumption 1** (quadratic growth). $\exists \alpha > 0$ *such that* $\forall \mathbf{x} \in \mathcal{P}$: *dist*$(\mathbf{x}, \mathcal{X}^*)^2 \leq 2\alpha^{-1}(f(\mathbf{x}) - f^*)$.

**Theorem 1** (Hoffman's bound (see for instance [2, 9])). *Suppose $\mathcal{P} \subset \mathbb{R}^d$ is a convex and compact polytope and let $f(\mathbf{x})$ be of the form $f(\mathbf{x}) = g(\mathbf{A}\mathbf{x}) + \mathbf{b}^\top \mathbf{x}$, where $g : \mathbb{R}^m \to \mathbb{R}$ is $\alpha_g$-strongly convex, $\mathbf{A} \in \mathbb{R}^{m \times d}, \mathbf{b} \in \mathbb{R}^d$. Then, $f(\cdot)$ has the quadratic growth property with some parameter $\alpha > 0$ (which depends on $\alpha_g$, $\mathbf{A}$ and the geometry of the polytope $\mathcal{P}$, see for instance [2, 9]).*

In particular, the highly important case of $f(\mathbf{x}) = \frac{1}{2}\|\mathbf{A}\mathbf{x} - \mathbf{b}\|^2$, where $\mathbf{A}$ is not necessarily full row-rank, satisfies the quadratic growth property w.r.t. any convex and compact polytope.

## 2.1 Lower bound for Frank-Wolfe-type methods

We now prove our claim that already for very simple problems and even when the (unique) optimal solution is a vertex of the polytope (i.e., $\dim \mathcal{F}^* = 0$), any Frank-Wolfe-type method (which we define next), even with away-steps, must exhibit at least linear dependence on the dimension, in worst case.

**Definition 1** (Frank-Wolfe-type method). *An iterative algorithm for Problem (1) is a Frank-Wolfe-type method if on each iteration $t$, it performs a single call to the linear optimization oracle of $\mathcal{P}$ w.r.t. the point $\nabla f(\mathbf{x}_t)$, i.e., computes some $\mathbf{u}_t \in \arg\min_{\mathbf{v} \in \mathcal{V}} \mathbf{v}^\top \nabla f(\mathbf{x}_t)$, where $\mathbf{x}_t$ is the current iterate, and produces the next iterate $\mathbf{x}_{t+1}$ by taking some convex combination of the points in $\{\mathbf{x}_1, \mathbf{u}_1, \ldots, \mathbf{u}_t\}$, where $\mathbf{x}_1$ is the initialization point.*

We let $\mathcal{S}_d^\downarrow$ denote the down-closed unit simplex in $\mathbb{R}^d$, i.e., $\mathcal{S}_d^\downarrow := \{\mathbf{x} \in \mathbb{R}^d \mid \mathbf{x} \geq 0, \sum_{i=1}^d \mathbf{x}(i) \leq 1\}$.

**Theorem 2.** *Consider the optimization problem $\min_{\mathbf{x} \in \mathcal{S}_d^\downarrow}\{f(\mathbf{x}) := \frac{1}{2}\|\mathbf{x}\|^2\}$. Then, any Frank-Wolfe-type method, when initialized with some standard basis vector $\mathbf{e}_i$, $i \in [d]$, must perform in worst case $\Omega(d)$ steps to obtain approximation error lower than $1/d$.*

*Proof.* Clearly, the unique optimal solution is $\mathbf{x}^* = \mathbf{0}$ and $f(\mathbf{x}^*) = 0$. Consider now the iterates of some Frank-Wolfe-type method and recall that $\mathbf{x}_1 = \mathbf{e}_i$ for some $i \in [d]$. Observe now that for any iteration $t$ for which it holds that $\text{nnz}(\mathbf{x}_t) = \text{nnz}(\nabla f(\mathbf{x}_t)) < d$ it follows that a valid output for the linear optimization oracle is a standard basis vector $\mathbf{e}_j$ such that $\mathbf{x}_t(j) = 0$. Thus, before making $d$ calls to linear optimization oracle, all iterates must lie in $\text{conv}\{\mathbf{e}_1, \ldots, \mathbf{e}_d\}$ and hence for all $t \leq d$ we have $f(\mathbf{x}_t) - f^* \geq 1/d$. $\qquad\square$

## 2.2 Strict complementarity condition

We now formally present the strict complementarity condition, which matches the one assumed in the early works of Wolfe [29] and Guélat and Marcotte [17].

**Assumption 2** (strict complementarity). *There exist $\delta > 0$ such that for all $\mathbf{x}^* \in \mathcal{X}^*$ and $\mathbf{v} \in \mathcal{V}$: if $\mathbf{v} \in \mathcal{V} \setminus \mathcal{F}^*$ then $(\mathbf{v} - \mathbf{x}^*)^\top \nabla f(\mathbf{x}^*) \geq \delta$; otherwise, if $\mathbf{v} \in \mathcal{V} \cap \mathcal{F}^*$ then $(\mathbf{v} - \mathbf{x}^*)^\top \nabla f(\mathbf{x}^*) = 0$.*

To motive Assumption 2 in the context of optimization with sparse/low-dimensional models under noisy data, we bring the following theorem which states that if the strict complementarity condition holds then, even if instead of directly optimizing $f(\cdot)$ over the polytope $\mathcal{P}$, we only optimize a noisy version of it $\tilde{f}(\cdot)$, then as long as the noise level is controlled by the strict complementarity parameter $\delta$, the optimal face is preserved. That is, the optimal solutions to the perturbed problem all lie within the optimal face w.r.t. the original objective $f(\cdot)$.

**Theorem 3.** *[strict complementarity implies robustness] Let $f(\cdot), \tilde{f}(\cdot)$ be two $\beta$-smooth, convex functions with quadratic growth with parameter $\alpha > 0$ over the polytope $\mathcal{P}$. Suppose also that for all $\mathbf{x} \in \mathcal{P}$, $\|\nabla f(\mathbf{x}) - \nabla \tilde{f}(\mathbf{x})\| \leq \nu$. Let $\mathcal{F}^*$ and $\tilde{\mathcal{F}}^*$ be the optimal faces w.r.t. the objective functions $f(\cdot)$ and $\tilde{f}(\cdot)$, respectively, and suppose the strict complementarity condition (Assumption 2) holds w.r.t. function $f(\cdot)$ and face $\mathcal{F}^*$ with parameter $\delta > 0$. If $\nu < \frac{\delta}{D(1+2\beta/\alpha)}$ then $\tilde{\mathcal{F}}^* \subseteq \mathcal{F}^*$.*

*Proof.* Let $\mathcal{X}^*$ and $\tilde{\mathcal{X}}^*$ denote the sets of optimal solutions w.r.t. $f(\cdot)$ and $\tilde{f}(\cdot)$, respectively. Fix some $\tilde{\mathbf{x}}^* \in \tilde{\mathcal{X}}^*$ and let $\mathbf{x}^* \in \mathcal{X}^*$ be the point in $\mathcal{X}^*$ closest in Euclidean distance to $\tilde{\mathbf{x}}^*$. From the convexity of $\tilde{f}(\cdot)$ we have that

$$f(\mathbf{x}^*) - f(\tilde{\mathbf{x}}^*) \geq (\mathbf{x}^* - \tilde{\mathbf{x}}^*)^\top \nabla f(\tilde{\mathbf{x}}^*) = (\mathbf{x}^* - \tilde{\mathbf{x}}^*)^\top \nabla \tilde{f}(\tilde{\mathbf{x}}^*) + (\mathbf{x}^* - \tilde{\mathbf{x}}^*)^\top (\nabla f(\tilde{\mathbf{x}}^*) - \nabla \tilde{f}(\tilde{\mathbf{x}}^*))$$
$$\geq 0 - \|\tilde{\mathbf{x}}^* - \mathbf{x}^*\|\nu = -\|\tilde{\mathbf{x}}^* - \mathbf{x}^*\|\nu,$$

| dimension ($d$) | avg. recovery error | avg. strict complementarity parameter ($\delta$) |
|:---:|:---:|:---:|
| 400 | 0.0117 | 0.8594 |
| 600 | 0.0132 | 0.5090 |
| 1000 | 0.0128 | 0.4505 |
| 1200 | 0.0142 | 0.5176 |

Table 2: Recovering a random sparse vector in the unit simplex $\mathbf{x}_0 \in \mathcal{S}_d$ with nnz($\mathbf{x}_0$) = 5 from noisy measurements $\mathbf{b} = \mathbf{A}\mathbf{x}_0 + c\|\mathbf{A}\mathbf{x}_0\|\mathbf{v}$, where $\mathbf{A} \in \mathbb{R}^{m \times d}$ has i.i.d. standard Gaussian entries, $\mathbf{v}$ is a random unit vector and $c = 0.2$. We set $m = 125$. For recovery we solve $\mathbf{x}^* \in \arg\min_{\mathbf{x} \in \tau \mathcal{S}_d} \|\mathbf{A}\mathbf{x} - \mathbf{b}\|^2$, where $\tau = 0.7$ (we need to scale down the unit simplex to avoid fitting the noise). The recovery error is $\|\tau^{-1}\mathbf{x}^* - \mathbf{x}_0\|^2$. The results are averaged over 50 i.i.d. runs.

where the last inequality follows from the optimality of $\tilde{\mathbf{x}}^*$ w.r.t. $\tilde{f}(\cdot)$ and the Cauchy-Schwarz inequality. Using the above inequality and the quadratic growth of $f(\cdot)$ we have that $\|\tilde{\mathbf{x}}^* - \mathbf{x}^*\|^2 = \text{dist}(\tilde{\mathbf{x}}^*, \mathcal{X}^*)^2 \leq \frac{2}{\alpha}\left(f(\tilde{\mathbf{x}}^*) - f(\mathbf{x}^*)\right) \leq \frac{2}{\alpha}\|\tilde{\mathbf{x}}^* - \mathbf{x}^*\|\nu$. Thus, we have that $\|\tilde{\mathbf{x}}^* - \mathbf{x}^*\| \leq 2\nu/\alpha$. It thus follows that for any vertex $\mathbf{v} \in \mathcal{V} \setminus \mathcal{F}^*$,

$$
\begin{aligned}
(\mathbf{v} - \mathbf{x}^*)^\top \nabla \tilde{f}(\tilde{\mathbf{x}}^*) &= (\mathbf{v} - \mathbf{x}^*)^\top \nabla f(\mathbf{x}^*) + (\mathbf{v} - \mathbf{x}^*)^\top (\nabla \tilde{f}(\tilde{\mathbf{x}}^*) - \nabla \tilde{f}(\mathbf{x}^*)) \\
&\quad + (\mathbf{v} - \mathbf{x}^*)^\top (\nabla \tilde{f}(\mathbf{x}^*) - \nabla f(\mathbf{x}^*)) \\
&\geq (\mathbf{v} - \mathbf{x}^*)^\top \nabla f(\mathbf{x}^*) - \|\mathbf{v} - \mathbf{x}^*\| \left(\beta\|\tilde{\mathbf{x}}^* - \mathbf{x}^*\| + \|\nabla \tilde{f}(\mathbf{x}^*) - \nabla f(\mathbf{x}^*)\|\right) \\
&\geq (\mathbf{v} - \mathbf{x}^*)^\top \nabla f(\mathbf{x}^*) - D(2\nu\beta/\alpha + \nu) \geq \delta - D\nu(1 + 2\beta/\alpha),
\end{aligned}
$$

where the first inequality follows from the smoothness of $\tilde{f}(\cdot)$ and the Cauchy-Schwarz inequality, and the last inequality follows from the strict complementarity assumption. Thus, we have that whenever $\nu < \frac{\delta}{D(1+2\beta/\alpha)}$ it must hold that $\tilde{\mathbf{x}}^* \in \mathcal{F}^*$. Otherwise, due to the differentiability of $\tilde{f}(\cdot)$, moving arbitrarily small positive mass from a vertex $\mathbf{v} \in \mathcal{V} \setminus \mathcal{F}^*$ in the convex decomposition of $\tilde{\mathbf{x}}^*$ (such $\mathbf{v}$ must exist since $\tilde{\mathbf{x}}^* \notin \mathcal{F}^*$), to the point $\mathbf{x}^*$ will reduce the objective value w.r.t. $\tilde{f}(\cdot)$, hence contradicting the optimality of $\tilde{\mathbf{x}}^*$. Thus, we have proved that $\tilde{\mathcal{F}}^* \subseteq \mathcal{F}^*$. $\qquad\square$

To further motivate the strict complementarity assumption, in Table 2 we bring empirical evidence for a standard setup of recovering a sparse vector from (random) linear measurements. In particular it is observable that the strict complementarity parameter $\delta$ does not change substantially as the dimension $d$ grows. Hence, in such a setup it is potentially much preferable to obtain convergence rates that depend on $\delta$ and $\dim \mathcal{F}^*$ (which in this case corresponds to the number of non-zero entries in the optimal solution) rather than on the ambient dimension $d$.

## 3 Main Results

---
**Algorithm 1** Frank-Wolfe Algorithm with line-search

---
1: $\mathbf{x}_1 \leftarrow$ some arbitrary point in $\mathcal{P}$
2: **for** $t = 1, 2 \ldots$ **do**
3: $\quad \mathbf{u}_t \leftarrow \arg\min_{\mathbf{u} \in \mathcal{V}} \mathbf{u}^\top \nabla f(\mathbf{x}_t)$
4: $\quad \mathbf{x}_{t+1} \leftarrow (1 - \eta_t)\mathbf{x}_t + \eta_t \mathbf{u}_t$ where $\eta_t \leftarrow \arg\min_{\eta \in [0,1]} f((1 - \eta)\mathbf{x}_t + \eta \mathbf{u}_t)$
5: **end for**

---

Before we get to the main result, we first begin with a very simple result proving that if the optimal solution is a vertex and the strict complementarity condition holds, then the standard Frank-Wolfe method with line-search (Algorithm 1) finds the optimal solution within a finite number of iterations, without even requiring the objective to satisfy the quadratic growth property. Such a result was essentially already proved in [17], though they **did assume** strong convexity of the objective, and did not give explicit complexity analysis (i.e., only proved finiteness).

**Theorem 4.** *Suppose* $\mathcal{F}^* = \{\mathbf{x}^*\}$ *where* $\mathbf{x}^* \in \mathcal{V}$. *Then, under Assumption 2, and **without assuming** quadratic growth of* $f(\cdot)$, *Algorithm 1 finds the optimal solution in* $O(\beta D^2/\delta)$ *iterations.*

*Proof.* Suppose Algorithm 1 runs for $T$ iterations and that the final iterate satisfies $\mathbf{x}_T \neq \mathbf{x}^*$. In particular it follows that $\mathbf{x}_T \in \text{conv}(\mathcal{V} \setminus \{\mathbf{x}^*\})$. Thus, from the convexity of $f(\cdot)$ and Assumption 2 it follows that $f(\mathbf{x}_T) - f^* \geq (\mathbf{x}_T - \mathbf{x}^*)^\top \nabla f(\mathbf{x}^*) \geq \delta$. However, from the standard convergence result for the Frank-Wolfe method (see for instance [19]), it follows that after $T = O(\beta D^2/\delta)$ iterations, $f(\mathbf{x}_T) - f^* < \delta$. Thus, we have arrived at a contradiction. $\qquad\square$

We now turn to present and prove our main result. For this result we use the Frank-Wolfe variant with away-steps already suggested in [17] and revisited in [21] without further change. Only the analysis is new and based mostly on the ideas of [12].

---

**Algorithm 2** Frank-Wolfe Algorithm with away-steps and line-search (see also [17, 21])

---

1: $\mathbf{x}_1 \leftarrow$ some arbitrary vertex in $\mathcal{V}$
2: **for** $t = 1, 2 \ldots$ **do**
3:      let $\mathbf{x}_t = \sum_{i=1}^n \lambda_i \mathbf{v}_i$ be a convex decomposition of $\mathbf{x}_t$ to vertices in $\mathcal{V}$, i.e., $\{\mathbf{v}_1, \ldots, \mathbf{v}_n\} \subseteq \mathcal{V}$, $(\lambda_1, \ldots, \lambda_n) \in \mathcal{S}_n$ and $\forall i \in [n] : \lambda_i > 0$ {maintained explicitly throughout the run of the algorithm by tracking the vertices that enter and leave the decomposition}
4:      $\mathbf{u}_t \leftarrow \arg\min_{\mathbf{v} \in \mathcal{V}} \mathbf{v}^\top \nabla f(\mathbf{x}_t)$, $i_t \leftarrow \arg\max_{i \in [n]} \mathbf{v}_i^\top \nabla f(\mathbf{x}_t)$, $\mathbf{z}_t \leftarrow \mathbf{v}_{i_t}$
5:      **if** $(\mathbf{u}_t - \mathbf{x}_t)^\top \nabla f(\mathbf{x}_t) < (\mathbf{x}_t - \mathbf{z}_t)^\top \nabla f(\mathbf{x}_t)$ **then**
6:          $\mathbf{w}_t \leftarrow \mathbf{u}_t - \mathbf{x}_t$, $\eta_{\max} \leftarrow 1$ {FW direction}
7:      **else**
8:          $\mathbf{w}_t \leftarrow \mathbf{x}_t - \mathbf{z}_t$, $\eta_{\max} \leftarrow \lambda_{i_t}/(1 - \lambda_{i_t})$ {away direction}
9:      **end if**
10:     $\mathbf{x}_{t+1} \leftarrow \mathbf{x}_t + \eta_t \mathbf{w}_t$ where $\eta_t \leftarrow \arg\min_{\eta \in [0, \eta_{\max}]} f(\mathbf{x}_t + \eta \mathbf{w}_t)$
11: **end for**

---

**Theorem 5.** *[Main Theorem] Let $\{\mathbf{x}_t\}_{t \geq 1}$ be the sequence of iterates produced by Algorithm 2 and for all $t \geq 1$ denote $h_t = f(\mathbf{x}_t) - f^*$. Then,*

$$\forall t \geq 1 : \qquad h_t = O(\beta D^2/t). \tag{2}$$

*Moreover, under Assumptions 1, 2 and assuming $\dim \mathcal{F}^* > 0$[3], there exists $T_0 = O(\beta D^2/(\delta^2 \kappa))$, where $\kappa = O(\psi^{*2} \dim \mathcal{F}^*/(\alpha \xi^{*2}))$ and $\delta$ is as defined in Assumption 2, such that*

$$\forall t \geq 2T_0 : \qquad h_{t+1} \leq h_{T_0} \exp\left(-\min\{\frac{1}{4}, \frac{1}{\beta\kappa D^2}\}\frac{t - 2T_0}{2}\right). \tag{3}$$

*Finally, under Assumptions 1, 2 and assuming $\dim \mathcal{F}^* > 0$, there exists $T_1 = O\left(1 + \beta D^2/(\delta^2 \kappa) + (1 + \beta\kappa D^2)\log(\kappa\beta D/\alpha)\right)$, such that the iterates $\{\mathbf{x}_t\}_{t \geq T_1}$ all lie inside the optimal face $\mathcal{F}^*$, and*

$$\forall t \geq 2T_1 : \qquad h_{t+1} \leq h_{T_1} \exp\left(-\min\{\frac{1}{4}, \frac{1}{\beta\kappa D_{\mathcal{F}^*}^2}\}\frac{t - 2T_1}{2}\right). \tag{4}$$

Note that the linear rates in (3), (4) depend explicitly only on the dimension of the optimal face $\dim \mathcal{F}^*$ (through the parameter $\kappa$), but not on the ambient dimension $d$. That is, treating all other quantities as constants, the linear rate is of the form $\exp(-\Theta(t/\dim \mathcal{F}^*))$ and not $\exp(-\Theta(t/d))$ as in the previous works [10, 12, 21]. Also, the rate in (4) depends only on the diameter of the optimal face and not on that of the entire polytope. Such improved dependence can be significant since for many polytopes, the diameter of a face scales with its dimension (e.g., the hypercube $[0, 1]^d$). Finally, we note that none of the parameters in Theorem 5 are required to run Algorithm 2 due to the use of line-search.

Before proving the theorem we will need a simple observation and two lemmas. Lemma 2 is the main technical novel ingredient and improves upon its counterpart in [12] by replacing the explicit dependence on the dimension $d$ with dependence only on $\dim \mathcal{F}^*$ and an additional (typically) lower-order term which depends on the strict complementarity parameter $\delta$.

Following the terminology of [21] we refer to each step $t$ of Algorithm 2 on which the away direction was chosen and also $\eta_t = \eta_{\max}$ as a *drop step*, since in such a case one of the vertices in the

decomposition of the current iterate $\mathbf{x}_t$ is removed from the decomposition. We denote by $T_{drop}$ the number of iterations up to (and including) iteration $T$ that are drop steps. The following simple observation is highly important for the analysis of Algorithm 2 and was made in [21].

**Observation 1.** *Let $\mathbf{x} \in \mathcal{P}$ be given by an explicit convex combination of $k$ vertices and suppose that starting with the point $\mathbf{x}$, $T$ iterations of Algorithm 2 have been executed. Then, on these $T$ iterations it holds that $T_{drop} \leq (k + T)/2$.*

**Lemma 1.** *Algorithm 2 satisfies that for all $t \geq 1$, $f(\mathbf{x}_t) - f^* = O(\beta D^2/t)$.*

*Proof.* Fix some iteration $t$ of Algorithm 2 on which the away direction was chosen but it is not a drop step (i.e., $\eta_t < \eta_{\max}$). Due to the use of line-search and the convexity of $f(\cdot)$ it in particular follows that $f(\mathbf{x}_t + \eta_t \mathbf{w}_t) = \arg\min_{\eta \geq 0} f(\mathbf{x}_t + \eta \mathbf{w}_t)$. Thus, we have that on such iteration,

$$\forall \eta \in [0,1]: \quad f(\mathbf{x}_{t+1}) = f(\mathbf{x}_t + \eta_t \mathbf{w}_t) \leq f(\mathbf{x}_t + \eta \mathbf{w}_t) \underset{(a)}{\leq} f(\mathbf{x}_t) + \eta \mathbf{w}_t^\top \nabla f(\mathbf{x}_t) + \eta^2 \beta \|\mathbf{w}_t\|^2/2$$

$$\underset{(b)}{\leq} f(\mathbf{x}_t) + \eta (\mathbf{u}_t - \mathbf{x}_t)^\top \nabla f(\mathbf{x}_t) + \eta^2 \beta D^2/2,$$

where (a) follows from the smoothness of $f(\cdot)$ and (b) follows since the away direction was chosen and not the FW direction. The above bound is the standard error-reduction analysis for the standard Frank-Wolfe method with line-search (Algorithm 1). Thus, we have that any iteration of Algorithm 2, which is not a drop step, reduces the error by at least the amount the Frank-Wolfe method with line-search reduces in worst-case. Since drop steps do not increase the function value, the lemma follows directly from the convergence rate of the standard Frank-Wolfe method (i.e., $O(\beta D^2/t)$, see for instance [19]) and Observation 1. $\qquad\square$

**Lemma 2.** *Let $\mathbf{x} \in \mathcal{P}$ and write $\mathbf{x}$ as a convex combination of points in $\mathcal{V}$, i.e., $\mathbf{x} = \sum_{i=1}^n \lambda_i \mathbf{v}_i$ such that $\lambda_i > 0$ for all $i \in [n]$. Let $\mathbf{x}^* \in \mathcal{X}^*$ be the optimal solution closet in Euclidean distance to $\mathbf{x}$. Then, $\mathbf{x}^*$ can be written as a convex combination $\mathbf{x}^* = \sum_{i \in [n]}(\lambda_i - \Delta_i)\mathbf{v}_i + \sum_{i \in [n]} \Delta_i \mathbf{z}$, for some $\mathbf{z} \in \mathcal{F}^*$, $\Delta_i \in [0, \lambda_i]$, and $\sum_{i \in [n]} \Delta_i \leq \min\left\{1, \delta^{-1}\left(f(\mathbf{x}) - f^*\right) + \frac{\sqrt{2 \dim \mathcal{F}^*}\psi^*}{\xi^* \sqrt{\alpha}} \sqrt{f(\mathbf{x}) - f^*}\right\}$.*

*Proof.* Let us write $\mathbf{x}$ as $\mathbf{x} = \sum_{i \in S_1} \lambda_i \mathbf{v}_i + \sum_{j \in S_2} \lambda_j \mathbf{v}_j$, where $S_1 = \{i \in [n] : \mathbf{v}_i \in \mathcal{F}^*\}$ and $S_2 = [n] \setminus S_1$. Since $\mathbf{x}^* \in \mathcal{F}^*$, clearly it must hold that for all $i \in S_2$, $\Delta_i = \lambda_i$, and $\mathbf{z} \in \mathcal{F}^*$.

We begin by upper-bounding $\sum_{i \in S_2} \Delta_i$. From the convexity of $f(\cdot)$ it holds that

$$f(\mathbf{x}) - f(\mathbf{x}^*) \geq (\mathbf{x} - \mathbf{x}^*)^\top \nabla f(\mathbf{x}^*) = \sum_{i \in S_1} \lambda_i (\mathbf{v}_i - \mathbf{x}^*)^\top \nabla f(\mathbf{x}^*) + \sum_{i \in S_2} \lambda_i (\mathbf{v}_i - \mathbf{x}^*)^\top \nabla f(\mathbf{x}^*)$$

$$\underset{(a)}{\geq} \sum_{i \in S_2} \lambda_i (\mathbf{v}_i - \mathbf{x}^*)^\top \nabla f(\mathbf{x}^*) \underset{(b)}{\geq} \sum_{i \in S_2} \lambda_i \delta,$$

where (a) follows from the optimality of $\mathbf{x}^*$ and (b) follows from the strict complementarity assumption (Assumption 2). Since for all $i \in S_2$ we have $\Delta_i = \lambda_i$ we obtain the bound $\sum_{i \in S_2} \Delta_i \leq \delta^{-1}(f(\mathbf{x}) - f^*)$.

We now turn to upper-bound $\sum_{i \in S_1} \Delta_i$. For this we use a refinement of the argument introduced in [12]. Applying Lemma 5.3 from [12] we have that there is alway a choice for $\{\Delta_i\}_{i \in [n]}$ and $\mathbf{z}$ such that for all $i \in [n]$, if $\Delta_i > 0$ then there must exist an index $j_i$ such that $\mathbf{A}_2(j_i)^\top \mathbf{z} = \mathbf{b}_2(j_i)$ and $\mathbf{A}_2(j_i)^\top \mathbf{v}_i < \mathbf{b}_2(j_i)$. Let $C^*(\mathbf{z}) \subseteq [m_2]$ be a set of minimal cardinality such that i) for all $j \in C^*(\mathbf{z})$, $\mathbf{A}_2(j)^\top \mathbf{z} = \mathbf{b}_2(j)$, and ii) for all $i \in S_1$ with $\Delta_i > 0$ there exist $j_i \in C^*(\mathbf{z})$ for which $\mathbf{A}_2(j_i)^\top \mathbf{v}_i < \mathbf{b}_2(j_i)$. The minimal cardinality implies that the latter requirement cannot hold for any subset of $C^*(\mathbf{z})$. Thus, it must hold that $|C^*(\mathbf{z})| \leq \dim \mathcal{F}^*$. This is true since otherwise, by definition of $\dim \mathcal{F}^*$, there must exist some $j \in C^*(\mathbf{z})$ such that $\mathbf{A}_2(j)$ can be written as a linear combination of rows in $\mathbf{A}_1^*$ (which correspond to constraints that are satisfied by $\mathbf{z}$ and $\{\mathbf{v}_i\}_{i \in S_1}$ since they are in $\mathcal{F}^*$) and the rows of $\mathbf{A}_2$ indexed in $C^*(\mathbf{z}) \setminus \{j\}$. However, this means that $j$ is redundant in $C^*(\mathbf{z})$, since if some $\mathbf{v}_i$ with $\Delta_i > 0$ satisfies all constraints indexed in $C^*(\mathbf{z}) \setminus \{j\}$, then it must also satisfy $\mathbf{A}_2(j)^\top \mathbf{v}_i = \mathbf{b}_2(j)$, which contradicts the minimality of $C^*(\mathbf{z})$.

Let $\mathbf{A}_{2,\mathbf{z}} \in \mathbb{R}^{|C^*(\mathbf{z})| \times d}$ be the matrix $\mathbf{A}_2$ after deleting each row $j \notin C^*(\mathbf{z})$. It holds that

$$\|\mathbf{x}^* - \mathbf{x}\|^2 \geq \frac{1}{\|\mathbf{A}_{2,\mathbf{z}}\|^2} \|\mathbf{A}_{2,\mathbf{z}}(\mathbf{x}^* - \mathbf{x})\|^2 = \frac{1}{\|\mathbf{A}_{2,\mathbf{z}}\|^2} \|\mathbf{A}_{2,\mathbf{z}} \sum_{i \in [n]} \Delta_i(\mathbf{z} - \mathbf{v}_i)\|^2$$

$$\underset{(a)}{\geq} \psi^{*-2} \sum_{j \in C^*(\mathbf{z})} \Big(\sum_{i \in [n]} \Delta_i(\mathbf{b}_2(j) - \mathbf{A}_{2,\mathbf{z}}(j)^\top \mathbf{v}_i)\Big)^2 \geq \psi^{*-2} \sum_{j \in C^*(\mathbf{z})} \Big(\sum_{i \in S_1} \Delta_i(\mathbf{b}_2(j) - \mathbf{A}_{2,\mathbf{z}}(j)^\top \mathbf{v}_i)\Big)^2$$

$$\geq \frac{1}{\psi^{*2}|C^*(\mathbf{z})|} \Big(\sum_{j \in C_0^*(\mathbf{z})} \sum_{i \in S_1} \Delta_i(\mathbf{b}_2(j) - \mathbf{A}_{2,\mathbf{z}}(j)^\top \mathbf{v}_i)\Big)^2$$

$$\underset{(b)}{\geq} \frac{1}{\psi^{*2}|C^*(\mathbf{z})|} \Big(\sum_{i \in S_1} \Delta_i \xi^*\Big)^2 \geq \frac{\xi^{*2}}{\psi^{*2} \dim \mathcal{F}^*} \Big(\sum_{i \in S_1} \Delta_i\Big)^2,$$

where (a) holds since the rows of $\mathbf{A}_2$ indexed in $C^*(\mathbf{z})$ are also rows of $\mathbf{A}_2^*$ and by the definition of $\psi^*$, and (b) follows from the definition of $C^*(\mathbf{z})$ and $\xi^*$.

Finally, using the quadratic growth of $f(\cdot)$ we have that $\sum_{i \in S_1} \Delta_i \leq \frac{\sqrt{2 \dim \mathcal{F}^*} \psi^*}{\xi^* \sqrt{\alpha}} \sqrt{f(\mathbf{x}) - f^*}$. $\square$

*Proof of Theorem 5.* Result (2) follows immediately from Lemma 1. From this result it also follows that for some $T_0 = O(\beta D^2/(\delta^2 \kappa))$ it holds that for all $t \geq T_0$, $\sqrt{h_t} \leq \delta\sqrt{\tilde{\kappa}}$, for $\tilde{\kappa} := 2 \dim \mathcal{F}^* \psi^{*2}/(\xi^{*2}\alpha)$. Throughout the rest of the proof, for every iteration $t$ we let $\mathbf{x}_t^*$ denote the point in $\mathcal{X}^*$ closest in Euclidean distance to the iterate $\mathbf{x}_t$.

Consider now some iteration $t \geq T_0$ and write the convex decomposition of $\mathbf{x}_t$ as $\mathbf{x}_t = \sum_{i=1}^n \lambda_i \mathbf{v}_i$, $\{\mathbf{v}_i\}_{\in[n]} \subseteq \mathcal{V}$. Suppose without loss of generality that $\mathbf{v}_1, \ldots, \mathbf{v}_n$ are ordered such that $\mathbf{v}_1^\top \nabla f(\mathbf{x}_t) \geq \mathbf{v}_2^\top \nabla f(\mathbf{x}_t) \geq \cdots \geq \mathbf{v}_n^\top \nabla f(\mathbf{x}_t)$. Let $\Delta^{(t)} = \sum_{i=1}^n \Delta_i$ be the bound in Lemma 2 when applied w.r.t. the point $\mathbf{x}_t$. Let $n_0$ be the smallest integer such that $\sum_{i=1}^{n_0} \lambda_i \geq \Delta^{(t)}$ and consider the point $\mathbf{p}_t = \big(\lambda_{n_0} - \big(\Delta^{(t)} - \sum_{j=1}^{n_0-1} \lambda_j\big)\big)\mathbf{v}_{n_0} + \sum_{i=n_0+1}^n \lambda_i \mathbf{v}_i + \Delta^{(t)} \mathbf{u}_t$, where $\mathbf{u}_t$ is the output of the linear optimization oracle in Algorithm 2. Since $\mathbf{p}_t$ is obtained by replacing vertices in the decomposition of $\mathbf{x}_t$ with highest inner product with $\nabla f(\mathbf{x}_t)$, with the point $\mathbf{u}_t$ that minimizes the inner-product among all points in $\mathcal{P}$, overall shifting the distribution mass which corresponds to the bound in Lemma 2, we have that (see also Lemma 5.6 in [12]) $(\mathbf{p}_t - \mathbf{x}_t)^\top \nabla f(\mathbf{x}_t) \leq (\mathbf{x}_t^* - \mathbf{x}_t)^\top \nabla f(\mathbf{x}_t)$. On the other hand, taking $\Delta_i = \lambda_i$ for all $1 \leq i < n_0$, $\Delta_{n_0} = \Delta^{(t)} - \sum_{j=1}^{n_0-1} \Delta_j$, and $\Delta_i = 0$ for all $n_0 + 1 \leq i \leq n$, we have that

$$(\mathbf{p}_t - \mathbf{x}_t)^\top \nabla f(\mathbf{x}_t) = \sum_{i=1}^n \Delta_i(\mathbf{u}_t - \mathbf{v}_i)^\top \nabla f(\mathbf{x}_t) \underset{(a)}{\geq} \sum_{i=1}^n \Delta_i(\mathbf{u}_t - \mathbf{z}_t)^\top \nabla f(\mathbf{x}_t)$$

$$= \Delta^{(t)}(\mathbf{u}_t - \mathbf{x}_t)^\top \nabla f(\mathbf{x}_t) + \Delta^{(t)}(\mathbf{x}_t - \mathbf{z}_t)^\top \nabla f(\mathbf{x}_t) \underset{(b)}{\geq} 2\Delta^{(t)} \mathbf{w}_t^\top \nabla f(\mathbf{x}_t),$$

where (a) and (b) follow from the definitions of $\mathbf{z}_t, \mathbf{w}_t$ in Algorithm 2. Thus, we have that $\Delta^{(t)} \mathbf{w}_t^\top \nabla f(\mathbf{x}_t) \leq \frac{1}{2}(\mathbf{x}_t^* - \mathbf{x}_t)^\top \nabla f(\mathbf{x}_t) \leq -\frac{1}{2} h_t$, where the last inequality follows from the convexity of $f(\cdot)$. In particular, it follows that for any $\rho > 0$, whenever $\rho\Delta^{(t)} \leq 1$ and either the FW direction was chosen or the away direction was chosen with $\eta_t < \eta_{\max}$ (i.e., not a drop step) that,

$$f(\mathbf{x}_{t+1}) = f(\mathbf{x}_t + \eta_t \mathbf{w}_t) \underset{(a)}{=} \arg\min_{\eta \in [0,1]} f(\mathbf{x}_t + \eta \mathbf{w}_t) \leq f(\mathbf{x}_t + \rho\Delta^{(t)} \mathbf{w}_t)$$

$$\underset{(b)}{\leq} f(\mathbf{x}_t) + \rho\Delta^{(t)} \mathbf{w}_t^\top \nabla f(\mathbf{x}_t) + \frac{\rho^2 \Delta^{(t)2} \beta \|\mathbf{w}_t\|^2}{2} \underset{(c)}{\leq} f(\mathbf{x}_t) - \frac{\rho}{2} h_t + 2\rho^2 \beta D^2 \tilde{\kappa} h_t, \qquad (5)$$

where (a) follows from the use of line-search and the convexity of $f(\cdot)$, (b) follows from the smoothness of $f(\cdot)$, and (c) follows from plugging the upper-bound on $\Delta^{(t)} \mathbf{w}_t^\top \nabla f(\mathbf{x}_t)$, the bound on $\Delta^{(t)}$ from Lemma 2 (note $\sqrt{h_t} \leq \delta\sqrt{\tilde{\kappa}}$ for all $t \geq T_0$, and thus $\Delta^{(t)} \leq 2\sqrt{\tilde{\kappa} h_t}$), and the Euclidean diameter of $\mathcal{P}$. Thus, for $\rho = \min\{1, 1/(8\beta\tilde{\kappa}D^2)\}$ by subtracting $f^*$ from both sides of (5), we get that for any step $t \geq T_0$ which is not a drop step, $h_{t+1} \leq \big(1 - \min\{\frac{1}{4}, \frac{1}{32\beta D^2 \tilde{\kappa}}\}\big) h_t$.

From Observation 1 we have that since the convex decomposition of $\mathbf{x}_{T_0}$ is supported on at most $T_0$ vertices and since on any iteration the approximation error never increases, from the above analysis we have that for all $t \geq 2T_0$,

$$h_t \leq h_{T_0} \exp\Big(-\min\{\frac{1}{4}, \frac{1}{\beta\kappa D^2}\}\big((t - T_0) - \frac{T_0 + (t - T_0)}{2}\big)\Big) = h_{T_0} \exp\Big(-\min\{\frac{1}{4}, \frac{1}{\beta\kappa D^2}\} \frac{t - 2T_0}{2}\Big).$$

This proves the rate in (3).

Finally, we turn to prove (4). Here we rely on similar arguments to those used in [17]. Using (2) and (3) we have that for $\tilde{T}_1 = O\left(1 + \beta D^2/(\delta^2 \kappa) + (1 + \beta \kappa D^2)\log(\kappa \beta D/\alpha)\right)$ it holds that for all $t \geq \tilde{T}_1$, $h_t < \min\{\alpha \delta^2/(8\beta^2 D^2), \beta D^2/2\}$. We now observe that for all $t \geq \tilde{T}_1$, $\mathbf{v} \in \mathcal{V} \setminus \mathcal{F}^*$, $\mathbf{y} \in \mathcal{V} \cap \mathcal{F}^*$:

$$(\mathbf{v} - \mathbf{y})^\top \nabla f(\mathbf{x}_t) = (\mathbf{v} - \mathbf{y})^\top \nabla f(\mathbf{x}_t^*) + (\mathbf{v} - \mathbf{y})^\top (\nabla f(\mathbf{x}_t) - \nabla f(\mathbf{x}_t^*)) \underset{(a)}{\geq} \delta - D\beta \|\mathbf{x}_t - \mathbf{x}_t^*\|$$

$$\underset{(b)}{\geq} \delta - \sqrt{2}\beta D\sqrt{h_t}/\sqrt{\alpha} \underset{(c)}{>} \delta/2, \tag{6}$$

where (a) follows from Assumption 2, the smoothness of $f(\cdot)$ and the Cauchy-Schwarz inequality, (b) follows from the quadratic growth of $f(\cdot)$, and (c) follows from our choice of $\tilde{T}_1$. Using similar arguments we make an additional observation that for all $t \geq \tilde{T}_1$ and $\mathbf{v} \in \mathcal{V} \setminus \mathcal{F}^*$,

$$(\mathbf{v} - \mathbf{x}_t)^\top \nabla f(\mathbf{x}_t) = (\mathbf{v} - \mathbf{x}_t^*)^\top \nabla f(\mathbf{x}_t^*) + (\mathbf{v} - \mathbf{x}_t^*)^\top (\nabla f(\mathbf{x}_t) - \nabla f(\mathbf{x}_t^*)) + (\mathbf{x}_t^* - \mathbf{x}_t)^\top \nabla f(\mathbf{x}_t)$$

$$\underset{(a)}{\geq} \delta + (\mathbf{v} - \mathbf{x}_t^*)^\top (\nabla f(\mathbf{x}_t) - \nabla f(\mathbf{x}_t^*)) \geq \delta - D\beta \|\mathbf{x}_t - \mathbf{x}_t^*\| > \delta/2$$

$$\underset{(b)}{>} (\mathbf{x}_t - \mathbf{u}_t)^\top \nabla f(\mathbf{x}_t), \tag{7}$$

where (a) follows from Assumption 2 and since $(\mathbf{x}_t^* - \mathbf{x}_t)^\top \nabla f(\mathbf{x}_t) \leq 0$, and (b) follows since from Theorem 2 in [21] we have that for all $t \geq \tilde{T}_1$, $(\mathbf{x}_t - \mathbf{u}_t)^\top \nabla f(\mathbf{x}_t) \leq D\sqrt{2\beta h_t} < \delta/2$.

From (6) it follows that for all $t \geq \tilde{T}_1$ it must hold that $\mathbf{u}_t \in \mathcal{V} \cap \mathcal{F}^*$. From (7) it follows that on every iteration $t \geq \tilde{T}_1$, if the convex decomposition of $\mathbf{x}_t$ assigns weight to some vertex in $\mathcal{V} \setminus \mathcal{F}^*$, then on that iteration the away-step must be chosen (and not a FW-step). In particular, using (6) again it must follow that the away vertex $\mathbf{z}_t$ satisfies $\mathbf{z}_t \in \mathcal{V} \setminus \mathcal{F}^*$. Moreover, on any such iteration it must also hold that $\eta_t = \eta_{\max}$, i.e., a drop step is performed. To see why the latter is true, suppose this does not hold, i.e., $\eta_t < \eta_{\max}$. Then it must hold that $(\mathbf{z}_t - \mathbf{x}_t)^\top \nabla f(\mathbf{x}_{t+1}) \leq 0$, since otherwise we can reduce the objective more by again moving away from $\mathbf{z}_t$, i.e., the choice of $\eta_t$ was sub-optimal. However, it holds that

$$(\mathbf{z}_t - \mathbf{x}_t)^\top \nabla f(\mathbf{x}_{t+1}) \underset{(a)}{\geq} (\mathbf{z}_t - \mathbf{x}_t)^\top \nabla f(\mathbf{x}_{t+1}^*) - D\beta \|\mathbf{x}_{t+1} - \mathbf{x}_{t+1}^*\| \underset{(b)}{\geq} (\mathbf{z}_t - \mathbf{x}_t)^\top \nabla f(\mathbf{x}_{t+1}^*) - \frac{D\beta\sqrt{2h_t}}{\sqrt{\alpha}}$$

$$= (\mathbf{z}_t - \mathbf{x}_{t+1}^*)^\top \nabla f(\mathbf{x}_{t+1}^*) + (\mathbf{x}_{t+1}^* - \mathbf{x}_t)^\top \nabla f(\mathbf{x}_{t+1}^*) - D\beta\sqrt{2h_t}/\sqrt{\alpha}$$

$$\underset{(c)}{\geq} (\mathbf{z}_t - \mathbf{x}_{t+1}^*)^\top \nabla f(\mathbf{x}_{t+1}^*) - h_t - D\beta\sqrt{2h_t}/\sqrt{\alpha}$$

$$\underset{(d)}{\geq} (\mathbf{z}_t - \mathbf{x}_{t+1}^*)^\top \nabla f(\mathbf{x}_{t+1}^*) - 2D\beta\sqrt{2h_t}/\sqrt{\alpha} \underset{(e)}{\geq} \delta - 2D\beta\sqrt{2h_t}/\sqrt{\alpha} > 0,$$

where (a) follows from the smoothness of $f(\cdot)$ and the Cauchy-Schwarz inequality, (b) follows from the quadratic growth and since $h_{t+1} \leq h_t$, (c) follows from convexity of $f(\cdot)$, (d) follows using the bound on the dual-gap again (Theorem 2 in [21]): $h_t \leq (\mathbf{x}_t - \mathbf{u}_t)^\top \nabla f(\mathbf{x}_t) \leq D\sqrt{2\beta h_t} \leq D\beta\sqrt{2h_t}/\sqrt{\alpha}$, and (e) follows from Assumption 2 and since $\mathbf{z}_t \in \mathcal{V} \setminus \mathcal{F}^*$. Thus, it must hold that $\eta_t = \eta_{\max}$. Recalling that for all $t \geq \tilde{T}_1$ it holds that $\mathbf{u}_t \in \mathcal{V} \cap \mathcal{F}^*$, we have that starting from iteration $T_1 = 2\tilde{T}_1$ and onwards, all iterates must lie inside the optimal face $\mathcal{F}^*$. Now, the rate in (4) follows from the same analysis as in the proof of (3), but this time noticing that since all the iterates lie inside $\mathcal{F}^*$ and the linear optimization oracle also only returns points in $\mathcal{F}^*$, we can replace the bound $\|\mathbf{w}_t\| \leq D$ with the tighter bound $\|\mathbf{w}_t\| \leq D_{\mathcal{F}^*}$. $\qquad \square$

## 4  Discussion

The recently studied linearly-converging Frank-Wolfe variants for polytopes have attracted notable attention and demonstrated superior empirical performance on several applications. Our work reveals an inherent flaw in all popular variants (at least those applicable to general polytopes) showing the worst-case convergence rate may depend on the ambient dimension which cannot explain their superior performance (Theorem 2). Strict complementarity can potentially bridge this gap, at least in case the optimal face is low-dimensional (Theorem 5). Low-dimensionality is expected in several popular models and applications (e.g., compressed sensing), and we motivate strict complementarity by proving it implies robustness of the optimal face to (deterministic) perturbations (Theorem 3), which may be interpreted as a certain notion of "well-conditionedness" of the optimization problem.

## Acknowledgements

We would like to thank Alejandro Carderera for pointing out a mistake in the last part of the proof of Theorem 5 in an earlier version of this paper. This research was supported by the ISRAEL SCIENCE FOUNDATION (grant No. 1108/18).

## Broader Impact

Not applicable for this work.

## Footnotes

[1]While in [10, 12] the dependence on the dimension is explicit in the convergence rate presented, in [21] it comes from the so-called pyramidal-width parameter, which already for the simplest polytopes such as the unit simplex or the hypercube $[0, 1]^d$ causes the worst-case rate to depend linearly on the dimension.

[2]This is not surprising, since when initialized with a vertex of the polytope, these methods increase the dimension of the active face, i.e., the face in which the current iterate lies, by at most one on each iteration.

[3]If $\dim \mathcal{F}^* = 0$ then the unique optimal solution is a vertex and it is straightforward to show that the rate (2) implies the same finite convergence as in Theorem 4.

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
