[Reviews · NeurIPS 2020]

Review 1

Summary and Contributions: This paper is a follow-up on the recent works of Lacoste-Julien & Jaggi (2015) and Garber & Hazan (2016). These prior works presented “away-step Frank-Wolfe” variants for minimization of a smooth convex objective function over a polytope with provable linear rates when the objective function satisfies a quadratic growth condition. These rates, however, explicitly depend on the problem dimensions, and they fail to explain the good empirical performance when in large dimensions. This paper investigates the following fundamental question, copied from the lines 55-56 verbatim: “Can explicit dependence on the dimension be avoided when the set of optimal solutions lies on a low-dimensional face of the polytope?” Here is a summary of the contributions: This paper 1- presents a simple example which proves that the answer for the central question above is negative in general, 2- revisits the strict complementarity condition introduced in the early literature of the FW method (and abandoned in the modern literature) and relates this condition some notion of robustness, 3- gives a new analysis that gives linear rates that do not depend on the dimension “d” but only to the dimension “dimF*” of the face of the polytope containing the solution set. [update after author feedback] I kindly disagree with the authors’ comments on the presentation. The three reviewers are not a good representative of all audiences in NeurIPS since they are assigned for their specific expertise. Also, there is a big difference in terms of the presentation style of this paper and other theoretical papers appeared in NeurIPS in previous years referred to in the authors’ feedback. One of the main goals of these big ML conferences is the information exchange between different fields of ML, so the authors are expected to present the work to the broader community and discuss the impact from various perspectives. I still have considerations about the presentation. I raise my score to 6 but I strongly recommend the authors to address a broader audience in the revision if the paper gets accepted.

Strengths: The paper takes a step to close the gap between the theoretical guarantees and the empirical performance of away-step Frank-Wolfe methods in large-scale problems. To my knowledge, the results are novel. The paper is well-written, and the analysis is easy to follow.

Weaknesses: For this work, the potential audience in the NeurIPS community is limited. The paper presents new convergence guarantees for an existing algorithm to tighten the dimension dependence of the constants of the convergence rate under strict complementarity assumption. The algorithm is already known to guarantee linear convergence, and the strict complementarity assumption is not easy to verify a priori. I do not see a big practical impact of the new results, as it is not discussed in the paper. The technical impact is not discussed either. Can this observation lead to new results in other FW variants or settings?

Correctness: The claims look correct. I have read the paper carefully and did not encounter any issues.

Clarity: The paper is well written, in the sense the novelty and the contributions are very clear. The abstract is concise and it provides a complete summary of the work. Technical results are presented in a systematic and clear way. The ideas and observations behind the analysis are well presented. All these aspects simplify the reading. On the negative side, the presentation is unusually technical for machine learning venues. It lacks discussions on the impacts of the new results, especially from the practical perspective. The paper finishes right after the proof of the main result, which feels like the paper is incomplete. I recommend the authors to include a concluding section that summarizes the takeaways and discusses the potential impacts.

Relation to Prior Work: Relation to prior work and the contributions of this work is clearly discussed in the introduction. In particular, Table 1 compares this paper with the three most related results from the existing literature.

Reproducibility: Yes

Additional Feedback: - Table 1: Although delta and dimF* are not available a priori, it is possible to construct these bounds a posteriori. That would be interesting to see a comparison of the standard bound that depends on the dimension “d”, the new bound, and the actual performance in some realistic numerical experiments. - Theorem 2: This example is clearly related to the examples in Jaggi (2013, Lemma 3) and Lan (arxiv1309.5550v2, 2014, Theorem 1), with the modification of “down-closed” unit simplex to produce a problem where the solution is at a vertex. I think this connection should be mentioned. - Line 142-147: This paragraph needs polishing. - Line 155: From the convexity of f^tilde we have -> From the convexity of f and f^tilde we have - Line 178-179: I recommend adding a small discussion to introduce Theorem 5 here. - Typo line 209: solution closet in Euclidean distance -> solution CLOSEST in Euclidean distance - Typo line 264: while FORM Theorem 2 -> while FROM Theorem 2 - Line 276: I recommend adding a Conclusions section.


Review 2

Summary and Contributions: This paper shows that the convergence rate of *Frank-Wolfe algorithm with away-step and line search* can actually have a better constant dependency. Assume that the optimal point lies on a lower-dimensional face of the polytope. The authors show that the algorithm's convergence rate depends on the dimension of the face, which can be much smaller than $d$ (i.e. the dimension of the polytope).

Strengths: A new result regarding *Frank-Wolfe algorithm with away-step and line search* is presented in this paper. Previous linear-rate results are of the form \exp( - t / d), while this paper shows that the rate can be improved to \exp( - t / dim(F*)) under strict complementary condition [Wolfe 1970], where dim(F*) is the dimension of the face that contains the optimal solution which can be much smaller than $d$. I believe that the result is significant as it shows when the algorithm can be faster ---- the result improves the previous theoretical guarantees of the algorithm.

Weaknesses: I understand that this is a theory paper. But it will be better if a proof-of-concept experiment is provided.

Correctness: I read the proof and the analysis thoroughly. It seems sound and correct. Typos: line 200: "\eta \in [0,1]" but \eta_{\max} could be larger than 1 line 210: "x^* = \sum_{i \in [n] } (lambda_i - Delta_i) + ... " v_i is missing .

Clarity: yes.

Relation to Prior Work: yes.

Reproducibility: Yes

Additional Feedback: *** after rebuttal *** I've read the authors' rebuttal and I maintain my score.


Review 3

Summary and Contributions: This paper revisits a classic result from Wolfe's book and (1) shows that in general, the dependency on the dimension in the rate is that of the ambient space. This is already extremely important as the sparsity of the solution does not improve this dependency. (2) it proves that under the strict complementarity condition the dependency can be improved to that of the face where the solution lays. ------------- Post Rebuttal: I decided to keep my score but I would like the authors to add a conclusion section that is used to position their work and discuss the implications of their results for the neurips community. Note that I am not asking for more experiments (as answered in the rebuttal).

Strengths: The paper is crisp and well written. It addresses an important open problem in optimization, unifying Wolfe's conjecture with the modern linear convergence analysis. I believe this paper may have a significant practical relevance as it significantly improves the understanding of the FW linear convergence rates and bridges a gap between theoretical analysis and previous experimental evidence.

Weaknesses: These are not weaknesses but rather questions. 1) Is there a general relation between the strict complementarity, F*, and the pyramidal width? I understand it in the case of the simplex, I wonder if something can be said in general. 2) It would be useful to discuss some practical applications (for example in sparse recovery) and the implication of the analysis to those. In general, I found the paper would be stronger if better positioned wrt particular practical applications. 3) I found the motivation in the introduction with the low-rank factorization unnecessary given that the main result is about polytopes. If the result has implications for low-rank matrix factorization I would like to see them explicitly discussed.

Correctness: I believe so

Clarity: Yes

Relation to Prior Work: Yes

Reproducibility: Yes

Additional Feedback:

[Author Response · NeurIPS 2020]

We would like to thank all reviewers for their time and effort invested in reviewing our work and for the valuable feedback. We now turn to address each of the reviewers individual comments.

**Reviewer #1:**   Thank you for your comments, for finding our results novel and for considering the question studied in this paper to be "fundamental".

We do not share your feelings regarding the claim that "the potential audience in the NeurIPS community is limited". We believe the ML community is eager for theory that pushes our understanding of such fundamental methods which are highly popular in our community. We give two past concrete examples: recent works on both FW with away-steps over polytopes (Lacoste Julien and Jaggi 2015, Garber and Hazan 2013, 2016) and FW over strongly-convex sets (Garber and Hazan 2015) were theoretical papers which have generated quite notable further research within the ML community. We believe this is due to the simple fact that those works, as we believe this current one also, presented simple yet powerful improvements to our current understanding of this very popular method. Note that there are already very recent works exploring the connections between strict complementarity and more-efficient optimization [2, 1, 3]

It is important to note that there is no practical need to verify the strict complementarity property since we do not present a new algorithm and the algorithm is independent of it.

Regarding your comment "presentation is unusually technical for machine learning venues", we would like to point out that all three reviewers have seem to perfectly understand the setting, the current state-of-affairs and contributions of this work. Nevertheless, we will make an effort to add some more explanations and discussions regarding applications of the results to standard Frank-Wolfe setups.

Additional feedback: 1. We believe the example in Table 2 demonstrates exactly this quite nicely. We can see a standard sparse recovery setup in which the strict complementarity parameter does not change substantially with the dimension, and so the benefit of the new bound over the previous which depends on the dimension is quite clear. We will add an appropriate discussion to clarify and emphasize this.

2. We will comment on the connection of our bound to previous FW lower-bounds.

3. Typos - thank you for catching these!

4. We will positively consider adding a conclusion section.

**Reviewer #2:**   Thank you for you positive feedback and for for finding our results significant.

Regarding experiments, we have included a simple experiment to demonstrate the existence of substantial strict-complementarity in a classical sparse-recovery setting (Table 2 in the paper). This experiment also clearly shows the benefit of the new bound over the previous - the strict complementarity does not change substantially even though the dimension does. We will add an appropriate discussion to make this point clearer. Also, since the algorithm analyzed is not new and has been implemented in many recent papers on various applications, we do not see great importance for additional experiments, as our mission is mainly to better understand its fundamental convergence properties. Please also refer to our answer to Reviewer #1 (line 5).

Thanks you for catching these typos!

**Reviewer #3:**   Thank you for you positive feedback, for your high appreciation of our work and for finding our results significant.

1. We are not sure there is a clear connection between these quantities. The pyramidal is a geometric property of the polytope, while strict complementarity obviously involves also the objective function.

2. Sparse recovery and applications: please see our response to Reviewer #2.

3. This work in only relevant for polytopes. We mentioned low-rank models to give further example of models in which a certain notion of sparsity is desired, beyond simply entry-wise sparsity.

# References

[1] Lijun Ding, Jicong Fan, and Madeleine Udell. $k$ fw: A frank-wolfe style algorithm with stronger subproblem oracles. *arXiv preprint arXiv:2006.16142*, 2020.

[2] Lijun Ding, Yingjie Fei, Qiantong Xu, and Chengrun Yang. Spectral frank-wolfe algorithm: Strict complementarity and linear convergence. *arXiv preprint arXiv:2006.01719*, 2020.

[3] Dan Garber. Linear convergence of frank-wolfe for rank-one matrix recovery without strong convexity. *arXiv preprint arXiv:1912.01467*, 2019.


[Meta-Review · NeurIPS 2020]

All reviewers agreed that this paper presented a strong theoretical contribution to NeurIPS by providing the first global convergence analysis of a projection-free method on generic polytopes which depends only on the dimensionality of the optimal face rather than the ambient dimension like previous analysis. Given the significant amount of practical and theoretical interest that Frank-Wolfe variants have had in the machine learning literature, this result is quite significant. The authors also provided a nice motivation for the addition of the strict complementary condition by showing a \Omega(d) lower bound on Frank-Wolfe variants when it does not hold. R1 was originally negative about the write-up as they thought that its (somewhat dry style) was lacking appeal to the broader community, and its technical impact was not sufficiently described. I have also read the paper in details and discussed it with the reviewers. I think the introduction and Section 2 did a decent job to motivate the results in the paper and present a broader appeal. While Section 3 is indeed on the technical side and the paper was lacking a discussion / conclusion, the reviewers agreed that this could be easily fixed in the camera ready version (and also thanks to the 9th page). I also appreciated the fact that the main paper is self-contained without the need for an appendix, unlike the recent trend on technical papers. R1 has thus upgraded their recommendation to 6, and I recommend the paper for acceptance. After my detailed read, I give further suggestions for the camera ready below. The authors should carefully implement the suggestions from the reviewers in the camera ready version, in particular adding a broader discussion about their Section 3 in conclusion / discussion, as well as providing further intuition / discussion of the strict complementary condition. == Other detailed comments == - L57 and elsewhere in the paper -- I agree with R3 that the several mentions of the low-rank condition in this paper is confusing (and slightly misleading) given that all the results only apply to polytopes and not the matrix cones where these low-rank conditions are applicable. I suggest that either it is removed from the paper; or only mentioned once (for context) with a clear caveat that this paper does not provide any theoretical guarantees for these kinds of constraint set. - L79 - I suggest to edit with something along the lines of "See also a follow-up work by B and Z [1] which later extended this result, but still had a dependence on the ambient dimension for generic polytopes." (Given that [1] has "general polytopes" in their title and that their abstract (somewhat misleadingly) claim that "similar rates" to Garber & Meshi [13] "can be generalized to arbitrary polytopes", one could easily assume that the nice dependence on the dimension of the optimal face from [13] could hold in [1] as well, so it is useful to be quite clear that this is not the case after all. - Table 1 is a bit confusing given that [1] *does give* a rate for general polytopes for a decomposition invariant version of AFW with line-search (AFW-2 in their paper), with a better rate for the unit simplex. I think for clarity and proper coverage of the related work, it would good to add a row for [1] with their generic polytope rate applied (naively) to the unit simplex (actually, I'd suggest to just put 3 rows in the table as the rate for [12],[20], naive [1] are the same here). Then in the caption, say something like "Illustrative comparison of assumptions and convergence rates over the unit simplex" (...) "For [1], we used their rate for generic polytopes applied to the unit simplex for illustration; they also provide the tighter rate with dim(F*) (like [13] for a different algorithm), but these rates do not apply to generic polytopes." Or, alternatively, it would be much more interesting to provide an example of a simple polytopes where the rates would be different and expressible in simple form (though I understand that this is not easy given that the pyramidal width [20] / facial distance [25] is not that easy to compute, so sticking to the unit simplex is fine).